# Holistically Evaluating the Environmental Impact of Creating Language Models

**Jacob Morrison**[1]    **Clara Na**[2]    **Jared Fernandez**[2]
**Tim Dettmers**[1,2]    **Emma Strubell**[1,2]    **Jesse Dodge**[1]

[1]Allen Institute for AI    [2]Carnegie Mellon University

jacobm@allenai.org

## Abstract

As the performance of artificial intelligence systems has dramatically increased, so too has the environmental impact of creating these systems. While many model developers release estimates of the power consumption and carbon emissions from the final training runs for their latest models, there is comparatively little transparency into the impact of model development, hardware manufacturing, and total water usage throughout. In this work, we estimate the real-world environmental impact of developing a series of language models, ranging from 20 million to 13 billion active parameters, trained on up to 5.6 trillion tokens each. When accounting for hardware manufacturing, model development, and our final training runs, we find that our series of models released **493 metric tons** of carbon emissions, equivalent to powering about 98 homes in the United States for one year, and consumed **2.769 million liters of water**, equivalent to about 24.5 years of water usage by a person in the United States, even though our data center is extremely water-efficient. We measure and report the environmental impact of our model development; to the best of our knowledge we are the first to do so for LLMs, and we find that model development, the impact of which is generally not disclosed by most model developers, amounted to ~**50%** of that of training. By looking at detailed time series data for power consumption, we also find that power usage throughout training is not consistent, fluctuating between ~15% and ~85% of our hardware's maximum power draw, with negative implications for grid-scale planning as demand continues to grow. We close with a discussion on the continued difficulty of estimating the environmental impact of AI systems, and key takeaways for model developers and the public at large.

## 1 Introduction

In recent years, the field of artificial intelligence has progressed at an unprecedented pace, driven in large part by the development and deployment of large language and multimodal models. However, the development of these models comes with significant environmental costs (Schwartz et al., 2020; Strubell et al., 2020; Wu et al., 2022). Training these models requires massive computational resources, which, in turn, require large amounts of energy. Powering training both emits carbon (by burning fossil fuels) and consumes water (by evaporating or polluting it in power plants, data centers, and hardware manufacturing processes; Li et al. (2023)). There is a growing demand for energy to power AI workloads, with projections estimating that datacenters may consume upwards of 11.7% of the total US energy demand by 2030 (Shehabi et al., 2024; Green et al., 2024). These energy needs are substantial such that they affect the decisions of both machine learning developers and energy providers – for instance, Microsoft recently signed a deal to purchase the next 20 years of energy generated by reopening a nuclear power plant,[1] and meanwhile energy providers are extending the life of aging fossil fuel energy plants to keep up with demand.[2] As such, especially as

---

[1]https://www.technologyreview.com/2024/09/26/1104516/three-mile-island-microsoft/
[2]https://www.wsj.com/business/energy-oil/electricity-demand-coal-gas-retirement-charts-dd07029a

increasing numbers of stakeholders become involved in the development and use of AI systems, it is imperative to carefully characterize the true cost of building and deploying state-of-the-art models, to inform effective strategies for mitigating potential harms and planning for future demand.

In this paper, we estimate the energy use and environmental impacts caused by training the OLMo series of transformer language models (Groeneveld et al., 2024; OLMo et al., 2025), ranging in size from 20 million to 13 billion active parameters, trained on 1.7 to 5.6 trillion tokens. To do this, we calculate Scope 2 $CO_2$ emissions in accordance with the Greenhouse Gas Protocol's definitions,[3] and Scope 1 and 2 water consumption following Li et al. (2023); in addition, we calculate "upstream" embodied carbon and water consumption, and provide "downstream" estimates from use of our models (which are part, but not all, of Scope 3).

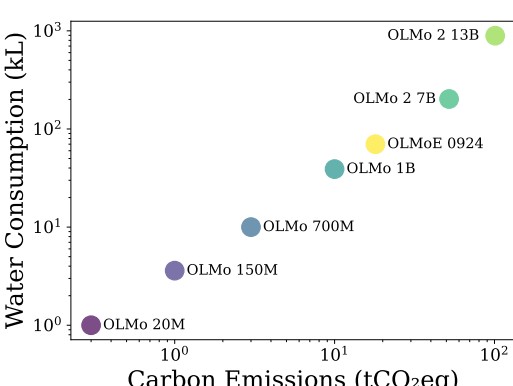

Figure 1: Environmental impact for a selection of the final training runs described in Section 4.1, where we rank each model by both its total water consumption and its $CO_2$ emissions. Our small models (<1B parameters) were trained on 1.7 trillion tokens, OLMo 1B was trained on 3 trillion, OLMo 2 7B was trained on 4 trillion, OLMoE was trained on 5 trillion, and OLMo 2 13B was trained on 5.6 trillion. We see that the total environmental impact for larger training runs is quite high, and increases quickly with model and dataset size.

Importantly, we calculate (i) electricity consumption, (ii) carbon emissions, and (iii) water consumption at three points in the machine learning pipeline: early model development (e.g., hyperparameter tuning and experiments before the final training run), training of the main model, and inference. To the best of our knowledge, we are the first to report this information for model development of large language models, and we find the environmental impact of developing even our relatively small models (only up to 13B parameters) is equivalent to burning 2.1 gasoline tanker trucks of fuel, or the amount of water consumed by one average person in the United States in about 7.5 years. We encourage the reader to consider larger models released by other organizations to have equivalently larger environmental impacts.

Our methodology draws upon best practices from recent publications, aiming to provide the most thorough reporting yet of the environmental impact of LLMs. For example, unlike previous works that assume GPUs operate at 100% of their theoretical maximum power draw (Dubey et al., 2024) and report only the cost to train a small set of released models, we measure power consumption at sub-second intervals throughout training. We focus our efforts on a wide range of model sizes, optimized for widespread deployment (Dubey et al., 2024; Mehta et al., 2024; Gemma Team et al., 2024), and estimate what the environmental impact would be if our models were deployed in a variety of different scenarios. We find that in some scenarios, our models would need to run inference on a few billion instances to match the electricity consumed, carbon emitted, and water consumed of the *entire* training process, a figure that can be reached by production systems in weeks to months based on current usage trends.[4]

We conclude that more transparency is needed across the industry in reporting the environmental impact of AI systems. Systems orders of magnitude larger than those in this paper are being built, and deployed at a global scale, leading to emissions 10s or 100s of times larger than what we report. This work is a step in the right direction, but responsibility of reporting and reducing the environmental impact must fall on those training the largest models, as they have the largest impact.

## 2 RELATED WORK

While most publicly available models do not report any climate impact, including $CO_2$ emissions, water usage, or embodied carbon, a few reports recently have included some estimates. For example,

---

[3] https://ghgprotocol.org/sites/default/files/standards/ghg-protocol-revised.pdf
[4] https://www.cnbc.com/2025/02/20/openai-tops-400-million-users-despite-deepseeks-emergence.html

Luccioni et al. (2023) reported estimates for emissions from the manufacturing process (embodied emissions), from electricity consumption during training, and from electricity consumption of the cluster while it was idle (see their Table 2). Dodge et al. (2022) measured electricity consumption and carbon emissions for training language models and computer vision models with granular timesteps with region-specific carbon intensity, but did not measure development costs, water consumption, or inference. Similarly, developers of the Llama models (Touvron et al., 2023a;b; Dubey et al., 2024) reported electricity consumption and carbon emissions estimates of training their final models; they did not estimate development cost or water consumption, and their approach to carbon intensity varied.[5] Gemma developers (Gemma Team et al., 2024) only report a single number: the total emissions from pretraining their models, not broken down by model or by different stages of training, or by electricity consumption and carbon intensity. The OLMo report (Groeneveld et al., 2024) documents electricity consumption per model, and uses region-specific carbon intensity to estimate emissions for two regions, but does not estimate other environmental impacts. The OLMo 2 report (OLMo et al., 2025) again documents electricity consumption per model and uses region- and datacenter-specific intensity factors to estimate emissions and also water consumption, but does not measure development costs or potential inference costs. Energy use and environmental impacts are not typically documented for proprietary models.

Comparably little transparency has been provided on the water consumption of AI systems. Li et al. (2023) estimate the water consumption of some closed models like GPT-3, but these estimates are based on speculation about location of training, energy consumption, etc., as there is very little public information about GPT-3's training. Similarly, there are few estimates of embodied carbon for AI systems, as the manufacturing process is notoriously opaque. In addition, almost all reporting of environmental impact is based on *training* of the *final* model that is released. Instead of only focusing on training, Luccioni et al. (2024) estimate the impact of inference of deployed AI systems. To the best of our knowledge our work provides the first public estimates of environmental impact of development of an LLM, i.e. hyperparameter tuning and ablations before the main training run.

## 3 METHODOLOGY

Our goal in this work is to characterize the holistic environmental impacts of large language models in as much detail as possible, enabling assessment of key challenges and future directions towards reducing those impacts. Typically, studies documenting language model training and development methodology will address this concern by reporting the cost to train the final, deployed model measured in GPU hours, kWh energy, and/or $CO_2$ emissions. However, this calculation provides an incomplete characterization of the factors leading to environmental degradation due to LLMs that under-estimates impacts and provides insufficient information to inform strategies for developing and deploying LLMs in a more environmentally conscious way.

Following the more comprehensive analysis provided for the BLOOM model (Luccioni et al., 2023), we expand our measurement to include both *operational* GHG emissions arising from the energy required for the development, training, and inference phases of the ML model lifecycle, as well as *embodied* emissions attributed to manufacturing of the hardware supporting those operations. We also go beyond previous work to report non-GHG externalities such as water use, and finer-grained data such as variance in energy use throughout training. We describe our methodology for measuring and estimating these impacts in more detail below.

### 3.1 OPERATIONAL IMPACTS

Operational environmental impacts of LLMs are those that arise directly from the development and use of models, and include the GHG emissions arising from energy sources used to power model training and deployment, including servers and data center cooling. We base our analysis of operational emissions around the following equation introduced by Schwartz et al. (2020) to describe the amount of computation required to produce a machine learning artifact, such as an LLM:

$$Cost(R) \propto E \cdot D \cdot H \tag{1}$$

---

[5]Llama 1 did not use the data center location's carbon intensity, instead using US national average carbon intensity; Llama 2 did not specify the carbon intensity; Llama 3 used a region-specific carbon intensity. All 3 assumed 100% GPU power draw throughout training.

where the cost of a scientific result $R$ (e.g. a claim that a particular training setup reaches $X$ accuracy on benchmark $Y$) is proportional to the product of the cost of processing a single example $E$, the size of the training dataset $D$, and the number of hyperparameter experiments $H$. In previous work, $E \cdot D$, the cost of training on the training dataset, is what is most commonly reported, and $H$, the total number of experiments, is most often excluded.

In our analysis, we calculate the total power consumption during model training, development, and inference, and use this to estimate the total carbon emissions and water consumption during each stage. We follow previous work (Luccioni et al., 2023; Dubey et al., 2024; Gemma Team et al., 2024) to calculate $CO_2$ emissions ($CO_2e$) from power consumption:

$$CO_2e = P \cdot PUE \cdot CI \tag{2}$$

where the total carbon emissions is equal to the power usage $P$, multiplied by the power usage effectiveness ($PUE$)[6] of the data center, multiplied by the carbon intensity $CI$ of the local power grid. We run all experiments in our two GPU clusters, Jupiter and Augusta, which are located in Texas and Iowa, respectively (see OLMo et al. (2025) for more information). Our 13B model was trained on Augusta, and all other experiments analyzed in this paper were trained on Jupiter.

Our data center providers informed us that Jupiter's $PUE$ is between 1.1 and 1.2 depending on the current total utilization (we conservatively assume 1.2 for our calculations), and that Augusta's trailing twelve-month average was 1.12. Jupiter is powered by Austin Energy, which most recently reported a carbon intensity of 0.332 kg $CO_2$ per kWh.[7] Augusta is located in Iowa, and the state of Iowa has an average carbon intensity of 0.352 kg $CO_2$ per kWh,[8] which we use for our calculations.

We follow Li et al. (2023) to calculate water consumed onsite and through power generation:

$$\text{Consumption} = P \cdot PUE \cdot (WUE_{\text{onsite}} + WUE_{\text{offsite}}) \tag{3}$$

where $WUE_{\text{onsite}}$ is the water usage effectiveness of the data center, dictated by the cooling hardware used, and $WUE_{\text{offsite}}$ is the water usage effectiveness of the local power provider, dictated by the precise mixture of sources of power generation, as thermo- and hydro-electric power plants lead to evaporated water that is lost and will not re-enter circulation in the local environment.

As our data center uses an efficient closed-loop cooling system with no evaporative cooling, we assume a $WUE_{\text{onsite}}$ of 0 liters per kWh. Following Reig et al. (2020), we assume a $WUE_{\text{offsite}}$ of 1.29 L per kWh for our Jupiter cluster and 3.10 L per kWh for our Augusta cluster.

Both calculations rely on total power usage. To calculate power usage during development and training, we analyze detailed time series data for a single node throughout each run, logging power data at sub-second intervals, and extrapolate to the total number of nodes. As we only measure GPU power consumption, our estimates should be viewed as a lower bound on the true amount of power consumed during development and training.

### 3.2 EMBODIED IMPACTS

Embodied impacts are those arising from the production of physical elements required to support LLM development and use, such as hardware manufacturing and data center construction. To calculate embodied emissions, we follow Luccioni et al. (2023) by amortizing the carbon emissions from manufacturing over the lifetime of the hardware to get an estimate of the per hour cost, and multiplying by the number of GPU hours used throughout model development and training. We extend this to include water consumption as well, by amortizing estimates of water consumption during manufacturing over the lifetime of the hardware.

---

[6] https://www.techtarget.com/searchdatacenter/definition/power-usage-effectiveness-PUE

[7] austinenergy.com/-/media/project/websites/austinenergy/commercial/carbonemissionscalculator.pdf

[8] www.eia.gov/electricity/state/iowa

### 3.3 Models, Data, and Hardware

Most of the models we evaluate are standard dense transformers, with an architecture similar to Llama (Touvron et al., 2023a;b; Dubey et al., 2024), OLMo (Groeneveld et al., 2024), and other recent popular models, ranging in size from 20 million to 13 billion active parameters. Each of the sub-billion parameter models was trained on 1.7 trillion tokens, the 1 billion parameter model was trained to 3 trillion tokens, the 7 billion parameter models were trained to 2, 3 and 4 trillion tokens, and the 13 billion parameter model to 5.6 trillion tokens. We additionally evaluate a mixture-of-experts (MoE) model with 1 billion active and 7 billion total parameters, trained to 5 trillion tokens.

Each model was trained on standard HGX servers with 8 NVIDIA H100 GPUs per server, with high speed interconnect between each node, and between 2 and 128 nodes concurrently per training run. All models except the 13B were trained in the same data center. See OLMo et al. (2025) for more information on our technical infrastructure.

### 3.4 Simulating Inference

Because we do not deploy our models, we do not collect or report data about real usage of our models. We instead report estimated costs associated with deployment of a subset of our models, along with comparison models, with varying inference configurations. In reality, causal language models can have a variety of use cases and be deployed on a variety of hardware infrastructure. As a representative deployment setting, we assume a setting in which users interact with the models via chat; we collect measurements assuming models are served on a single H100 GPU via SGLang (Zheng et al., 2024). All three inference configurations used can be mapped to a previously proposed realistic online inference scenario (Reddi et al., 2020; Peng et al., 2023). Specifically, other than the "batching" scenario where all requests are sent instantaneously, the requests follow a Poisson distribution, albeit at different rates that influence different batch sizes. The requests themselves come from the ShareGPT dataset,[9] and each inference scenario involves the same sample of 2400 prompts (same random seed). Input and output lengths, therefore, are the same in theory for a given model, but due to differences in tokenization and model context length, there are slight variations in mean input/output lengths across models, 225-250 and 190-230 tokens respectively.

In our inference experiments, we measure cumulative energy consumption using CodeCarbon (Courty et al., 2024) tracking, which was verified against the same time series monitoring used throughout training. Notably, we measure total power and energy consumption associated with only the relevant processes, excluding the overhead associated with, for example, holding the model in memory or listening for requests.

We ran our inference simulations on our Jupiter cluster, used to train almost all of our models, but we use only a single H100 GPU at a time. See Appendix A.1 for details about our inference methodology and assumptions.

## 4 Results

### 4.1 Building Our Models

In this section we aim to report a full accounting of the environmental impact of training our series of models, from hardware manufacturing, to development, and the final training runs. We follow the methodology outlined in Section 3.1 and Section 3.2.

When calculating environmental impact, we use information from our data center providers and their power providers to measure the efficiency of each cluster. For Jupiter, the cluster used to train all models but the 13B, we assume a carbon intensity of 0.332 kg $CO_2$ emitted per kWh, a power usage effectiveness (*PUE*) of 1.2, and a total water usage effectiveness (*WUE*) of 1.29 liters per kWh. For Augusta, the cluster used to train the 13B, we assume a carbon intensity of 0.351 kg $CO_2$ emitted per kWh, a PUE of 1.12, and a total WUE of 3.1 liters per kWh.

---

[9]https://huggingface.co/datasets/anon8231489123/ShareGPT_Vicuna_unfiltered/resolve/main/ShareGPT_V3_unfiltered_cleaned_split.json,anon8231489123/ShareGPT_Vicuna_unfiltered

Table 1: We developed our models in five groups, based on parameter count and architecture: less than 1 billion, 1 billion, 7 billion, and 13 billion parameters, and our mixture-of-experts model with 1 billion active and 7 billion total parameters. We found that ∼70% of our developmental environmental impact came from developing the 7B and 13B models, and the total impact was emissions equivalent to 2.1 tanker trucks' worth of gasoline, and equal to about 7 and a half years of water used by the average person in the United States.

| | GPU Hours | Total MWh | # Runs | Carbon Emissions (tCO$_2$eq) | Equivalent to... (energy usage, 1 home, U.S.) | Water Consumption (kL) | Equivalent to... (water usage, 1 person) |
|---|---|---|---|---|---|---|---|
| **<1B** | 29k | 19 | 20 | 6 | 1 yr, 4 mo | 24 | 3 mo |
| **7B** | 269k | 196 | 375 | 65 | 13 yrs, 6 mo | 252 | 2 yrs, 7 mo |
| **13B** | 191k | 116 | 156 | 46 | 9 yrs, 7 mo | 402 | 3 yrs, 7 mo |
| **MoE** | 27k | 19 | 35 | 6 | 1 yr, 4 mo | 24 | 3 mo |
| **Total** | 680k | 459 | 813 | 159 | 33 yrs, 1 mo | 843 | 7 yrs, 5 mo |

**Hardware manufacturing**   NVIDIA does not release the embodied carbon emissions or water consumption about the hardware it produces, so we assume the same embodied carbon emissions as Luccioni et al. (2023), or 3700 kg of CO$_2$eq per 8x server node, equal 463 kg per GPU. There is little public information on how much water is required to produce a single GPU, though chip manufacturing facilities require millions of liters per day.[10] Some estimates[11] place TSMC water usage at 12.33 liters per square centimeter of hardware, which equals 100.4 liters per H100, which we use for our analysis.

We additionally estimate the environmental impact from mining rare earth metals used during manufacturing, assuming an H100 is 0.1% rare earth metal by mass. Mining 1 kg of rare earth materials consumes about 11 kL of water and releases 65.4 kg CO$_2$eq (Browning et al., 2016), and one 12-inch silicon wafer weighs 125 grams[12] and produces about 63 H100s.[13] [14] Together, these add an additional 2.2 liters consumed and 0.013 kg CO$_2$eq per GPU.

Internally, we assume a 4 year lifespan for our GPUs, which leads to an embodied emissions of 0.013 kg of CO$_2$eq and 0.003 liters of water consumed per GPU hour when the estimated embodied impacts is amortized over the assumed lifetime of the GPU. We used 1.65 million GPU hours in total, leading to a total of **22 tCO$_2$eq** emitted and **4.8 kL** of water consumed during manufacturing.

**Development**   Before launching our final training runs for each model, we ran a series of controlled experiments to stabilize and improve our training setup, to explore different parameter initializations and mid-training recipes, and to determine our final hyperparameters and data mixtures through scaling law experiments (Bhagia et al., 2024). We ran these in five distinct groups: small models with less than 1 billion parameters, 1 billion parameter models, 7 billion parameter models, 13 billion parameter models, and our mixture-of-experts model. We report detailed development costs for each group in Table 1.

Unsurprisingly, we find that the majority of development costs (∼70%) were incurred at the 7 and 13 billion parameter scale, due to both the relative size of the model and our own prioritization, and we see this both in the total environmental impact and the number of individual runs per category. Using our data center's efficiency factors, we find that our development runs led to **159 tCO$_2$eq** emitted and **843 kL** of water consumed.

**Final training runs**   Finally, we fully trained our series of models, ranging from 20 million to 13 billion active parameters, with detailed information provided in Table 2. As we saw during development, the majority of the cost incurred came from training our 7B and 13B models, which we trained to 2 to 5 trillion tokens. We also see that the 1B dense model required about as much energy per trillion tokens as the MoE model with 1B active parameters, though the MoE model was slightly less efficient, most likely due to the extra compute required for routing tokens. In summary, we find that our training runs led to **312 tCO$_2$eq** emitted and **1,921 kL** of water consumed.

---

[10] https://www.azcentral.com/story/opinion/op-ed/joannaallhands/2024/06/12/tsmc-arizona-water-use-recycling/74059522007/

[11] https://www.semiconductor-digest.com/water-supply-challenges-for-the-semiconductor-industry/

[12] https://web.archive.org/web/20131207002716/http://wafercare.com/Page.aspx?id=1012

[13] https://anysilicon.com/die-per-wafer-formula-free-calculators/

[14] https://developer.nvidia.com/blog/nvidia-hopper-architecture-in-depth/

Table 2: We list the estimated power usage, carbon emissions, and water consumption from training our dense transformers, ranging from 20 million to 13 billion parameters, trained on 1.7 to 5.6 trillion tokens, and a mixture-of-experts model with 1 billion active and 7 billion total parameters, trained to 5 trillion tokens. We find that the environmental impact is quite high, even for our relatively small models. Training our series of models emitted equivalent carbon to over 65 years of electricity use by the average household in the U.S., and consumed equivalent water to the average person in the U.S. for about 17 years.
\* One of the original OLMo 7B models was trained on LUMI, which runs entirely on hydroelectric power. See Groeneveld et al. (2024) for more information.
† denotes unreleased models that were trained for various internal experiments.

| | Power Usage (MWh) | Carbon Emissions ($tCO_2eq$) | Equiv. to... (energy usage, 1 home, U.S.) | Water Consumption (kL) | Equiv. to... (water usage, 1 person, U.S.) |
|---|---|---|---|---|---|
| **Gemma 2B & 9B** | - | 131 | 25 yrs, 11 mo | - | - |
| **Llama 2 7B** | 81 | 31 | 6 yrs, 1 mo | - | - |
| **Llama 2 13B** | 162 | 62 | 12 yrs, 2 mo | - | - |
| **Llama 3.1 8B** | - | 420 | 83 years | - | - |
| **Llama 3.2 1B** | - | 107 | 14 years | - | - |
| **OLMo 20M†** | 0.8 | 0.3 | 3 weeks | 1 | 3 days |
| **OLMo 60M†** | 1.2 | 0.4 | 1 month | 1.6 | 5 days |
| **OLMo 150M†** | 2.4 | 1 | 2 mo, 1 wk | 3.6 | 12 days |
| **OLMo 300M†** | 5 | 2 | 5 months | 5.9 | 19 days |
| **OLMo 700M†** | 8 | 3 | 7 months | 10 | 33 days |
| **OLMo 7B†** | 67 | 22 | 4 yrs, 4 mo | 87 | 9 months |
| **OLMo 1B (3T)** | 30 | 10 | 2 years | 39 | 4 months |
| **OLMo 7B** | 149 | 0* | - | 0* | - |
| **OLMo 7B (Twin)** | 114 | 70 | 13 yrs, 10 mo | 487 | 4 yrs, 4 mo |
| **OLMo (04\|07)24 7B** | 95 | 32 | 6 yrs, 4 mo | 122 | 1 yr, 1 mo |
| **OLMo 2 7B** | 157 | 52 | 10 yrs, 4 mo | 202 | 1 yr, 9 mo |
| **OLMo 2 13B** | 230 | 101 | 21 years | 892 | 7 yrs, 10 mo |
| **OLMoE 0924** | 54 | 18 | 3 yrs, 7 mo | 70 | 7 months |
| **Total (Ours)** | 913 | 312 | 65 years | 1,921 | 17 yrs, 1 mo |

**Putting it in perspective** In total, our series of models led to at least **493 $tCO_2eq$** emitted. Using the U.S. Environmental Protection Agency's Greenhouse Gas Equivalencies Calculator[15], this is equivalent to 6.5 tanker trucks' worth of gasoline burned, emissions from the average yearly energy use for 98.2 homes in the U.S., or the amount of carbon sequestered by 472 acres of U.S. forests in one year. We additionally estimate we consumed at least **2,769 kL** of water, which is equivalent to about 24 and a half years of water consumption by the average person in the U.S.[16]

**Other Costs** In this work, we strive to provide a thorough accounting of the total cost of developing our models. However, there remain a number of sources of emissions and water consumption that are difficult, if not impossible to comprehensively measure without access to proprietary information across a range of industries, such as transportation and end of life hardware disposal. While the costs we report above represent a large portion of the total development process, more transparency is needed to understand the full impact of model training.

## 4.2 SIMULATING DEPLOYMENT & INFERENCE

We report *simulated* inference costs; that is, we explore the question of what our models' impact might be if they were put into production. In contrast to §4.1, where we reported the actual impact from our actions, this section reports partial estimates of Scope 3 carbon emissions and water consumption: the impact from the downstream actions of others using our models. We include comparisons with recent instruction-tuned models as well.

In Table 3, we display 1) power and energy costs, 2) carbon and water consumption, and 3) the time to complete 100 requests. We additionally report "breakeven" points, that is the number of

---

[15]https://www.epa.gov/energy/greenhouse-gas-equivalencies-calculator
[16]https://www.epa.gov/watersense/statistics-and-facts

Table 3: Measurements and estimates of resource costs from SGLang benchmarking on 2400 prompts from ShareGPT at varying request rates. Since the models were served on machines from the same cluster that our OLMo 2 models were trained on, we use the same WUE and PUE coefficients of 1.29 L / kWh and 1.2 respectively, and carbon intensity of 0.332 kg $CO_2$e / kWh. Note the difference in units for energy consumption and carbon emissions, namely MWh $\rightarrow$ kWh, tons $\rightarrow$ grams $CO_2$eq, and kL $\rightarrow$ L. The measurements reported in this table account for the GPU processes associated with active inference, but not CPU or RAM associated with e.g. server overhead. Thus, these numbers can be considered as lower bounds on usage in similar settings. Also of note is the relatively small variability in carbon emissions and water consumption across different model sizes in cases where batches are not saturated, despite faster inference in smaller models when fully saturated; greater peak efficiency does not guarantee efficient deployment if inference is not optimized. We do not report "break-even" points for Qwen 2.5 because its training costs are not public.

| Model | Request freq. (req / s) | GPU Power Usage (kWh) | Carbon Emissions (g $CO_2$eq) | Water consump. (L) | Seconds per 100 req. | # Inf. for $CO_2$ equiv. w/ training |
|---|---|---|---|---|---|---|
| Llama 3.2 1B | $\infty$ | 0.003 | 1.0 | 0.004 | 1.38 | 258 bil. |
| | 8 | 0.036 | 12.0 | 0.054 | 12.64 | 21.5 bil. |
| | 1 | 0.160 | 53.1 | 0.238 | 100.58 | 4.83 bil. |
| Qwen 2.5 7B | $\infty$ | 0.009 | 3.0 | 0.013 | 1.79 | — |
| | 8 | 0.053 | 17.6 | 0.079 | 12.77 | — |
| | 1 | 0.308 | 102.3 | 0.459 | 100.58 | — |
| Llama 3.1 8B | $\infty$ | 0.011 | 3.7 | 0.016 | 2.13 | 276 bil. |
| | 8 | 0.051 | 16.9 | 0.076 | 12.79 | 59.5 bil. |
| | 1 | 0.333 | 110.6 | 0.496 | 100.64 | 9.12 bil. |
| Llama 2 13B | $\infty$ | 0.034 | 11.3 | 0.051 | 6.53 | 13.3 bil. |
| | 8 | 0.060 | 19.9 | 0.089 | 13.09 | 7.52 bil. |
| | 1 | 0.401 | 133.1 | 0.597 | 100.73 | 1.13 bil. |
| OLMo 1 1B (3T) | $\infty$ | 0.004 | 1.3 | 0.006 | 0.99 | 18.2 bil. |
| | 8 | 0.038 | 12.6 | 0.057 | 12.63 | 1.91 bil. |
| | 1 | 0.165 | 54.8 | 0.246 | 100.58 | 441 mil. |
| OLMo 2 7B | $\infty$ | 0.018 | 6.0 | 0.027 | 3.68 | 20.9 bil. |
| | 8 | 0.049 | 16.3 | 0.073 | 12.88 | 7.68 bil. |
| | 1 | 0.358 | 118.9 | 0.533 | 100.54 | 1.05 bil. |
| OLMo 2 13B | $\infty$ | 0.033 | 11.0 | 0.049 | 6.60 | 22.1 bil. |
| | 8 | 0.057 | 18.9 | 0.085 | 13.05 | 12.8 bil. |
| | 1 | 0.386 | 128.2 | 0.575 | 100.57 | 1.89 bil. |
| OLMoE 0924 | $\infty$ | 0.006 | 2.0 | 0.009 | 1.70 | 21.7 bil. |
| | 8 | 0.037 | 12.3 | 0.055 | 12.82 | 3.51 bil. |
| | 1 | 0.151 | 50.1 | 0.225 | 100.60 | 861 mil. |

inferences in each scenario required for inference costs to be equal or greater to training costs. See Table 4 in Appendix A.1 for additional results.

We find that for most models tested, the number of inferences required to outweigh training costs is in the hundreds of millions to tens of billions, except for the most over-trained models. As many of these models were created to be efficient in deployment-focused scenarios – such as on edge devices, or in popular online products – it is important to consider inference costs in addition to training costs. The largest model providers are producing up to hundreds of billions of tokens per day,[17] highlighting that deployed models can quickly reach this tipping point.

## 4.3 POWER FLUCTUATIONS DURING TRAINING

One problem caused by training AI models at large scales is that the power demand starts and stops suddenly (Dubey et al., 2024), which power grids can struggle to handle. When demand sharply rises, generation sources that can be quickly started and stopped – generally powered by fossil fuels, such as coal and natural gas – must be brought online quickly, increasing the marginal carbon intensity of the grid and potentially negatively impacting other consumers in cases where demand rises more quickly than generation can handle. When demand sharply drops, excess power is discarded– by grounding the power or venting steam–until generation sources can spin down. Power grids can generally manage some large variations (for example, when communities experience a sudden

---

[17] https://x.com/sama/status/1756089361609981993

power outage), but as we add more variability to the system, it becomes more difficult to maintain this delicate balance, and infrastructure is not set up to handle frequent, large fluctuations.

In Figure 2, we show a snapshot of our model's GPU power consumption during pre-training. We find that power consumption is not consistent – instead, power is consistent *while the model is training*, but drops quickly while saving checkpoints. Though our models are relatively small, and we have since improved checkpointing performance, other model developers have experienced similar issues caused by checkpointing and synchronization between nodes (Dubey et al., 2024).

## 5 DISCUSSION

### 5.1 MORE TRANSPARENCY IS (STILL) NEEDED

While many model developers–including some of the largest for-profit entities operating in this space–make best efforts to report at least part of the cost of building their AI systems (Dubey et al., 2024; Gemma Team et al., 2024), more transparency is still needed throughout the development pipeline. The EU AI Act,[18] and some proposed legislation, such as the Artificial Intelligence Environmental Impacts Act[19] in the United States, would start the process for defining voluntary environmental impact reporting standards for model developers, but until such standards are widespread in the community, improved transparency can only come through voluntary efforts by companies and research organizations. Policy action is needed to ensure there is public visibility into environmental impacts across the entire supply chain, from hardware manufacturing, data center construction, and energy production, all the way through to model deployment and inference.

**Embodied emissions are still an enigma** Though a vital piece of all model development pipelines, the environmental impact of manufacturing the GPUs used is essentially unknown. In previous work, Wu et al. (2022) and Luccioni et al. (2023) highlighted the fact that researchers focused on AI's environmental impact are forced to use unreliable estimates of the cost of manufacturing state-of-the-art computational hardware, and the situation is no better now, nearly two years later. Many companies that manufacture other pieces of data center hardware disclose estimates of the lifetime environmental impact,[20] and until GPU manufacturers release similar information–on a voluntary or compulsory basis–this will not improve.

**Development costs are substantial, and unreported** As reported in Section 4.1, we present detailed information on the cost of developing our training pipeline, in contrast with previous work. We found that development costs–associated with failed runs, hyperparameter searches, testing architecture changes, and more–are responsible for a substantial portion of the total environmental impact of creating our systems, highlighting a need for more transparency from developers. This is especially important in light of AutoML tools, where many models may be automatically trained while searching for a solution, and scaling law experiments, where smaller models are trained to predict the performance of larger models, and then discarded (Li et al., 2024; Bhagia et al., 2024).

**Water costs are real, and under-explored** While under-explored in previous work, AI's growing water consumption is beginning to receive more and more attention[21] (Li et al., 2023), though not as much as it may deserve. As shown in Section 4.1, even training a series of comparatively small models uses a large amount of water, the amount of which is also drastically impacted by both the cooling systems used in data centers as well as the power generation methods used. Without more transparency from developers on when, where, and how they are training their models, it will continue to be difficult to quantify the scale of the issue, stymieing efforts to address it.

### 5.2 SMALL CHOICES DURING TRAINING CAN HAVE LARGE IMPACTS

While many issues relating to transparency require action from corporations and large research groups, choices made during training have a large effect downstream.

---

[18] https://artificialintelligenceact.eu/article/95/
[19] https://www.markey.senate.gov/imo/media/doc/artificial_intelligence_environmental_impacts_act_of_2024_-_020124pdf.pdf
[20] https://www.hpe.com/psnow/doc/a50005151enw
[21] https://www.washingtonpost.com/technology/2024/09/18/energy-ai-use-electricity-water-data-centers/

**Smaller models are cheaper to train and use, but at what cost?** Until recently, to achieve high model performance, a large model was needed. Compute-optimal scaling laws for neural network training (Hoffmann et al., 2022; Kaplan et al., 2020) imply that it is more efficient to put more data into a larger model, because of diminishing returns from "overtraining" a small model. This meant that models were expensive to both train and deploy, limiting how widespread they could become, and how financially feasible they were to be used in a variety of scenarios.

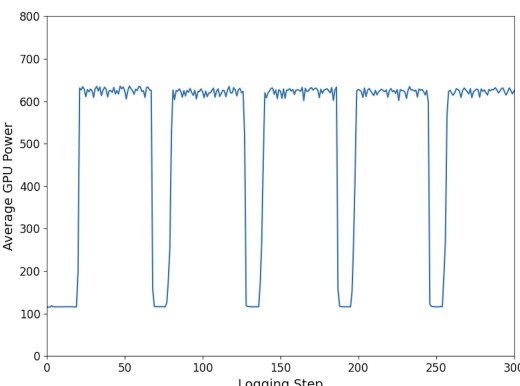

Figure 2: Average GPU power for a single node for the first 300 logging steps during OLMo 2 7B training. The first spike is the beginning of training, and each drop happens when a model checkpoint is saved. When actively training, the average GPU power is over 600W, over 85% of an H100's maximum power draw of 700W, and during checkpointing, power usage drops to just over 100W, or about 15% maximum.

Recently, however, continuing to train models on more and more tokens beyond the "compute-optimal" limit[22] has been extremely successful in making "deployment-optimized" models that can be substantially cheaper to perform inference with. This has led to an explosion in both training cost for small models, and total inference compute cost, as API-based models become cheaper to use[23][24] and small models are deployed on-device (Gunter et al., 2024; Abdin et al., 2024). This may be an instance of Jevons' Paradox (Jevons, 1865): when a resource's efficiency increases, overall consumption of that resource tends to increase, rather than decrease. In other words, as the cost of training models decreases, the downstream impact may continue to grow.

This is especially clear in context of our results in Section 4.2, showing that though the raw number of inferences required to outweigh training is objectively quite large, smaller models are being deployed in many new scenarios that will drastically increase their total usage. Many inference use cases are also not able to be batched (e.g. generating text on a phone for immediate use), meaning that deployers cannot schedule these requests to take advantage of cheaper or cleaner energy, and must make use of immediately available power. Given that this trend will most likely only accelerate, it is vital that we improve transparency into the total cost of deployment in all scenarios.

**Power fluctuations reveal inefficiencies at best, challenges to power grid control at worst** While it is known that the dramatic spike in power consumption at the beginning of training and the subsequent drop at the end are problematic for power grid operators at large scales, little has been discussed publicly about how power consumption changes throughout training. We found that our models, using an optimized code base and publicly available tooling, sees rapid power fluctuations throughout training caused by the commonplace practice of frequently saving model checkpoints. This means that without careful engineering, one training run can cause thousands of rapid power fluctuations, which poses an immediate challenge for large-scale LLM training in data centers, which typically source energy directly from power providers. Generated power needs to go somewhere, and rapid, large drops in consumption during training breaks common assumptions about data center supply and demand, leading to significant control challenges in power systems. While some frameworks have begun to implement workarounds to manage this issue,[25] more awareness is needed on the part of researchers and engineers as training runs scale to tens of thousands of GPUs[26] or more, as even some of the largest model developers encounter difficulties from regularly shifting power demand throughout training due to checkpointing, awaiting collective communications, and other unforeseen and potentially catastrophic failures (Dubey et al., 2024). We emphasize that addressing this will require more comprehensive solutions such as parallelized checkpointing, improved demand response in data centers running large AI workloads, and new, heterogeneous methods for distributed training spanning software, hardware, and scheduling.

---

[22]e.g. scaling from 1 to 2 to 15T tokens for Llama 1, 2, and 3 (Touvron et al., 2023a;b; Dubey et al., 2024)

[23]https://openai.com/index/gpt-4o-mini-advancing-cost-efficient-intelligence/

[24]https://developers.googleblog.com/en/gemini-15-flash-updates-google-ai-studio-gemini-api/

[25]E.g. the new PYTORCH_NO_POWERPLANT_BLOWUP environment variable in PyTorch.

[26]https://time.com/7021709/elon-musk-xai-grok-memphis/

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

# A   APPENDIX

## A.1   ADDITIONAL INFERENCE SIMULATION DETAILS

We benchmark models using the ShareGPT dataset, assuming an online inference chat setting. In practice, with much longer inference examples, OLMo models may have an "unfair" advantage in that they were generally trained with context lengths shorter than the other models we benchmark. However, we do not believe that to be a significant factor in our results. In fact, we observe that Llama 3.1 8B is actually measured to be faster and less energy intensive than OLMo 7b models, likely due to the use of grouped-query attention (GQA; Ainslie et al. (2023)) in Llama 8b, vs not in OLMo models.

We report additional inference simulation results on a larger set of models in Table 4,

## A.2   LIMITATIONS

Our main limitations are discussed throughout the main body of this work – in particular, we make various assumptions about embodied impacts due to lack of real data, and our inference and deployment numbers were benchmarked in a controlled, limited setting, as we do not in reality serve our models in the same sense, and we do not have access to data about most other models' real usage.

We present only a limited set of inference simulations following a number of simplistic assumptions. Specifically, we simulate only settings where a deployed model is ingesting input tokens and generating output tokens following default parameters defined in SGLang (Zheng et al., 2024) – as opposed to, for instance, evaluating only the likelihood of a given text. Additionally, we note that practitioners frequently employ different inference-time optimizations such as quantization; perform generation with different decoding algorithms; and/or deploy to and run inference on edge devices, sometimes even without GPUs. We do not account for this variety of scenarios in our experiments.

We observe linear trends in training costs relative to parameter count across four orders of magnitude and eight model sizes. However, we do not necessarily expect that this trend would hold tightly in all training settings across all possible scales – for instance, decentralized training or training across multiple datacenters might be expected to incur significantly greater communication overhead throughout training. Though we have not trained these models ourselves, our hope is that our work will encourage others working in a broad range of settings to provide their own holistic reports of environmental resource consumption.

Table 4: Full version of Table 3 in §4.2. Measurements and estimates of resource costs from SGLang benchmarking on 2400 prompts from ShareGPT at varying request rates. The models were served on machines from the same cluster that our models were trained on, so we use the same WUE and PUE coefficients of 1.49 L / kWh and 1.2 respectively, and carbon intensity of 0.332 kg $CO_2$e / kWh. The measurements reported in this table account for the GPU processes associated with active inference, but not CPU or RAM associated with e.g. server overhead. Thus, these numbers can be considered as lower bounds on usage in similar settings. We do not report "break-even" points for Qwen models since the training costs are not public.

| | Request freq. (req / s) | GPU Power Usage (kWh) | Carbon Emissions (g $CO_2$eq) | Water consump. (L) | Seconds per 100 req. | # Inf. for $CO_2$ equiv. w/ training |
|---|---|---|---|---|---|---|
| Llama 3.2 1B | $\infty$ | 0.003 | 1.0 | 0.004 | 1.38 | 258 bil. |
| | 8 | 0.036 | 12.0 | 0.054 | 12.64 | 21.5 bil. |
| | 1 | 0.16 | 53.1 | 0.238 | 100.58 | 4.83 bil. |
| Llama 2 7B | $\infty$ | 0.019 | 6.3 | 0.028 | 3.58 | 11.9 bil. |
| | 8 | 0.054 | 17.9 | 0.08 | 12.83 | 4.18 bil. |
| | 1 | 0.349 | 115.9 | 0.52 | 100.62 | 647 mil. |
| Llama 3 8B | $\infty$ | 0.01 | 3.3 | 0.015 | 1.93 | 282 bil. |
| | 8 | 0.052 | 17.3 | 0.077 | 12.78 | 54.2 bil. |
| | 1 | 0.337 | 111.9 | 0.502 | 100.63 | 8.37 bil. |
| Llama 3.1 8B | $\infty$ | 0.011 | 3.7 | 0.016 | 2.13 | 276 bil. |
| | 8 | 0.051 | 16.9 | 0.076 | 12.79 | 59.5 bil. |
| | 1 | 0.333 | 110.6 | 0.496 | 100.64 | 9.12 bil. |
| Llama 2 13B | $\infty$ | 0.034 | 11.3 | 0.051 | 6.53 | 13.3 bil. |
| | 8 | 0.06 | 19.9 | 0.089 | 13.09 | 7.52 bil. |
| | 1 | 0.401 | 133.1 | 0.597 | 100.73 | 1.13 bil. |
| Qwen 2.5 1.5B | $\infty$ | 0.003 | 1.0 | 0.004 | 0.86 | – |
| | 8 | 0.033 | 11.0 | 0.049 | 12.65 | – |
| | 1 | 0.163 | 54.1 | 0.243 | 100.57 | – |
| Qwen 2.5 7B | $\infty$ | 0.009 | 3.0 | 0.013 | 1.79 | – |
| | 8 | 0.053 | 17.6 | 0.079 | 12.77 | – |
| | 1 | 0.308 | 102.3 | 0.459 | 100.58 | – |
| Qwen 2.5 14B | $\infty$ | 0.018 | 6.0 | 0.027 | 3.45 | – |
| | 8 | 0.058 | 19.3 | 0.086 | 13.02 | – |
| | 1 | 0.387 | 128.5 | 0.577 | 100.64 | – |
| Qwen 1.5 MoE (2.7BA, 14BT) | $\infty$ | 0.01 | 3.3 | 0.015 | 2.64 | – |
| | 8 | 0.043 | 14.3 | 0.064 | 13.11 | – |
| | 1 | 0.165 | 54.8 | 0.246 | 100.68 | – |
| OLMo 1 1B | $\infty$ | 0.004 | 1.3 | 0.006 | 0.99 | 18.2 bil. |
| | 8 | 0.038 | 12.6 | 0.057 | 12.63 | 1.91 bil. |
| | 1 | 0.165 | 54.8 | 0.246 | 100.58 | 441 mil. |
| OLMo 0724 7B | $\infty$ | 0.017 | 5.6 | 0.025 | 3.33 | 29.8 bil. |
| | 8 | 0.052 | 17.3 | 0.077 | 12.77 | 9.73 bil. |
| | 1 | 0.339 | 112.5 | 0.505 | 100.59 | 1.49 bil. |
| OLMo 2 7B | $\infty$ | 0.018 | 6.0 | 0.027 | 3.68 | 20.9 bil. |
| | 8 | 0.049 | 16.3 | 0.073 | 12.88 | 7.68 bil. |
| | 1 | 0.358 | 118.9 | 0.533 | 100.54 | 1.05 bil. |
| OLMo 2 13B | $\infty$ | 0.033 | 11.0 | 0.049 | 6.6 | 22.1 bil. |
| | 8 | 0.057 | 18.9 | 0.085 | 13.05 | 12.8 bil. |
| | 1 | 0.386 | 128.2 | 0.575 | 100.57 | 1.89 bil. |
| OLMoE 0924 (1BA, 7BT) | $\infty$ | 0.006 | 2.0 | 0.009 | 1.7 | 21.7 bil. |
| | 8 | 0.037 | 12.3 | 0.055 | 12.82 | 3.51 bil. |
| | 1 | 0.151 | 50.1 | 0.225 | 100.6 | 861 mil. |

