# OpenReview forum: "Holistically Evaluating the Environmental Impact of Creating Language Models"
_ICLR.cc/2025/Conference — ICLR 2025 Spotlight_

### Official Review · Reviewer_TYDT · 2024-11-02

**Soundness:** 3
**Presentation:** 3
**Contribution:** 3
**Rating:** 6
**Confidence:** 5

**Summary:**

This paper discusses the impact of large language models on the environment by studying the costs associated with training and deploying them. It explores the hidden cost of training a model, particularly when it comes to hardware manufacturing and the pre-training steps of creating a model, along with the training and deploying costs. They run their experiments on a small set of models with parameter sizes ranging between 20 million and 7 billion parameters. The results show that models released 270 metric tons of carbon emissions and 1.137 million liters of water. Additionally, the author discusses the power fluctuation during training that is a result of model checkpointing.

**Strengths:**

The discussion of energy consumption and costs incurred by training and deploying LLMs is a rising and highly relevant topic.
This paper has the following strengths:

1. It is one of the first studies to report the environmental impact of model development, not just the final training runs, highlighting the significant but often overlooked costs associated with hyperparameter tuning and other development activities.
2. Cost evaluation encompasses power consumption, carbon emissions, and water usage and is not confined to a single aspect.
3. Reporting the costs and putting them into perspective with real-world equivalencies.

**Weaknesses:**

Even though the topic is highly interesting he are some limitations to the paper:
1. Lack of novelty: The issue of power consumption in LLMs has been widely studied, and this paper doesn't provide any additional ideas, metrics, or insights expect for the study development cost.
2. The findings are based on a specific set of small models, which may limit the generalizability of the results to other models and data centers with different configurations and efficiencies
3. The study does not include data from actual deployment and usage of the models, relying instead on simulated scenarios, which may not fully reflect the actual environmental costs. In fact, the paper has a limited set of inference simulations with very simplistic assumptions, which may not fully capture the real-world deployment scenarios and their environmental impacts.
4. Some of the calculations rely on assumptions and estimates, particularly regarding the embodied emissions and water consumption of hardware manufacturing, which may not be entirely accurate.
5. Limited comparison across models: The authors seem to have taken the carbon consumption of llama and OLMo in Table 2 from previous works without replicating results, which meant no water usage comparison for training. For deployment, they only compare with Llama.
6. Given the small sizes of the models, the paper lacks an analysis of how their results scale to larger models.

**Questions:**

1. In table 2, why is OLMo reported twice is there a difference?
2. I am curious to know how much time it took to train your models. Given the hardware resources you had (up to 64 HGX servers with h100), why was your study limited to 7 billion parameter models?

---

> ### Author Response · Authors · 2024-11-25
>
> Thank you for reviewing our work, and for your suggestions.
>
> W1: Regarding novelty, we would like to push back on this point. We do not claim to have developed a novel methodology for calculating the environmental impact of training language models. Instead, we aim to set a new standard for reporting the total environmental impact, and encourage other developers in the community to meet this standard going forward. We aim to provide a comprehensive, holistic evaluation, in contrast with many recent technical reports that only evaluate carbon emissions, assume GPUs are always operating at 100% of their maximum power draw, and only report training costs. In other words, we aim to show that this level of detail is feasible to report, and we encourage others to do so as well.
>
> W2: Regarding model size and generalizability, we agree that different model architectures and training setups (hardware, data center locations, etc) would have an impact on the downstream environmental impact. However, the calculations we perform hold at all model sizes, and location-specific variables (PUE, WUE, etc) can be substituted as necessary.
>
> W3 is discussed in our manuscript in both the relevant methods and results subsections, 3.4 and 4.2. To reiterate: Regarding deployment in real world scenarios, we agree that our estimates of the cost of model deployment are limited in comparison to real-world data. However, as we state in the paper, we do not host our own models, and thus do not have access to real world data. Instead, our estimates aim to show potential impact, and we encourage those hosting models at large scales to share similar analyses with their own real-world data in the future. In general, we do not report our precise deployment simulation numbers as part of any central claim we make; instead, we include these results to contextualize the relative costs of training and deployment.
>
> W4 is also discussed in our manuscript (see 5.1, in our paragraph titled “Embodied emissions are still an enigma.”). To reiterate: Regarding embodied emissions, we agree that our estimates likely are not 100% accurate, as we state in the paper. Instead, we aim to provide a better estimate of the embodied emissions compared to previous work, and to highlight how little information regarding embodied emissions is publicly available, which we discuss in Section 5. Additionally, we make many efforts to obtain real information and estimates from our providers (including contacting our data center providers), and we make reasonable, conservative assumptions about information that we were not able to obtain. We aim to be very careful highlighting the aspects of our estimates that are based on assumptions vs. “real” data.
>
> W5: Regarding previously reported environmental impacts, can you explain more about what you mean by “replicating their results,” with regards to OLMo’s and Llama’s carbon emissions? They have not released power consumption data, and thus we must instead take their reported numbers at face value. You do raise a good point though, and we will include estimates of the water consumption *as if their models were trained in our data centers*, as we do not have access to location information for their training runs. In the final version, we will also add comparisons with other models in the deployment estimate section, such as Qwen 2.5.
>
> W6: Regarding model size, we disagree that 7 billion parameter models are not representative of the impact of training larger models. Especially with the growing popularity of deployment-optimized models (such as Gemini Flash, Claude Haiku, GPT-4o mini, etc), we believe that smaller models are only becoming more popular in deployment, especially for on-device settings. However, we have also recently completed training a 13 billion parameter model. We can report that the 13B model, trained to 4 trillion tokens, required about 290 MWh (vs ~157 for the 7B trained to 4T tokens), showing an almost exactly linear trend in training costs as model size grows. We are still calculating other costs for this model, but we will include the full results in the final version.
>
> To answer your questions:
> * In the OLMo paper (https://arxiv.org/pdf/2402.00838), they released two separate 7B models, and reported the carbon emissions from both models separately, as they were trained on different hardware in different clusters. We compare against both OLMo models in our paper, but we will make it more clear that these are separate models in the final version.
> * We would like to emphasize that training models at the 7B parameter scale and higher is very expensive, in terms of compute, environmental impact, and money. We are training our models to between 2 and 4 trillion tokens, so scaling beyond 7B is very expensive. However, as mentioned above, we have since trained a 13 billion parameter model, and we will include the environmental impact of training this model (also above) in the final version of the paper.

---

> > ### Comment · Reviewer_TYDT · 2024-11-25
> >
> > I thank the authors for their thorough responses and for addressing most of my concerns. While I appreciate the effort and clarity provided, I still have a couple of additional questions and suggestions for consideration:
> >
> > W1—Thank you again for clarifying your focus on setting a new reporting standard rather than developing a novel methodology. I appreciate the emphasis on providing a comprehensive evaluation and encouraging the community to adopt this level of detail in their analyses. That said, I wonder how you differentiate this work from being categorized as a detailed technical report for your private model, particularly since you do not claim methodological novelty, the equations used are based on prior works, and much of the data of other models rely on referencing other technical reports.
> >
> > W4— Indeed, it would be highly beneficial if you highlight and distinguish the real vs simulated data.
> >
> > W5— I understand that the previous environments did not report their power consumption data, and I appreciate your acknowledgment of this limitation. My point was primarily to highlight this gap. That said, I believe that retraining some of these models on X tokens under your settings, if feasible, could provide valuable estimates to address this gap and offer further insights into the potential environmental impacts under consistent conditions.
> >
> > Furthermore, if the authors include water consumption estimates based on their own data center conditions, it will be crucial to clearly distinguish these estimates in the manuscript to avoid any misinterpretation or association with the original reported results. This clarity will be key to maintaining accuracy. Additionally, I highly appreciate the inclusion of new models and look forward to seeing how they contribute to the broader context of your analysis.
> >
> > W6—Including results from your 13B parameter model in the final version will certainly bolster the generalizability of your claims about training costs across model sizes. It may also help to explicitly highlight this linear trend in training costs in the methods or results sections to strengthen your argument for scaling costs.

---

> > > ### Author Response · Authors · 2024-11-26
> > >
> > > W5: Retraining other models would be an infeasible experiment to add on to this paper, unfortunately. First, only OLMo has released their exact pretraining data, so replicating Llama’s pretraining is not possible, even ignoring potential access issues to similar training code, hardware, etc. Training other 7B models for even 1% of our maximum token count would cost roughly $50,000, based on conservative estimates of GPU hour market prices (https://www.hyperstack.cloud/gpu-pricing), which is too expensive for a single experiment for nearly all organizations. We instead compare many model sizes using the same architecture, trained to similar token counts, to show environmental impacts under otherwise consistent conditions.
> > >
> > > Additionally, running these experiments using our setup would only allow us to answer a single research question: how much impact does the architecture have? While this is an interesting research question, to appropriately answer it we would recommend significant experimentation beyond the models we compare against, and careful ablations changing each part of the architectures, which we leave to future work.
> > >
> > > W1: To clarify, we do claim novelty in our work: we are the first to report all of (i) electricity consumption, (ii) CO2 emissions, and (iii) water consumption at three points in the machine learning pipeline: early model development, training, and inference. At the time of submission, our work is the state of the art with respect to estimation of resource consumption from training language models.
> > >
> > > With respect to concerns that “*the equations used are based on prior works*”, we chose to use the equations we did because these are the current best practice methods for calculating the environmental impact of model training given the information we have. For example, to report CO2 emissions, our Equation 2 uses power consumption, PUE, and grid carbon intensity. This equation provides the best estimate with the data we have, and if we had changed it we would no longer have the best estimate of CO2 emissions. Our work uses best practice throughout, and thus we intend for it to act as a reference for future researchers on how to report on the environmental impact throughout the machine learning pipeline. We hope that this convinces the reviewer that using the correct equations for reporting the most relevant information is not a reason to decrease the score of the paper.
> > >
> > > > *detailed technical report for your private model*
> > >
> > > To be clear, our models are not private, and will be publicly released by the time of final publication. We have withheld identifying information in order to preserve anonymity, but we will provide links to all of our models in the final version of the paper.
> > >
> > > To reiterate, as the reviewer acknowledges in the followup response, we believe that one of the key contributions of our work is to set a new standard for reporting of resource consumption during training of language models. Our hope is that our work will prompt others to report with a similar or greater amount of detail and care, and we believe it meets the standards for novelty and impact for a conference publication.
> > >
> > > W4: To be clear, we already do work to distinguish between real and simulated data (e.g. Section 4.2, *Simulating Deployment & Inference*), but we will make sure that there is no confusion throughout. We’re also happy to provide extra clarity now if the reviewer feels there are any specific parts of the manuscript that are unclear.
> > >
> > > W6: Thank you for the suggestion. To reiterate, we have already established a linear trend across three orders of magnitude, with seven model sizes. We will include the 4th order of magnitude/8th model size (the 13B model), but we believe that our results stand on their own already.
> > >
> > > More generally, the reviewer mentioned clarity multiple times. Is there a particular part of the submission that they feel could have been written more clearly, other than the meaning of the 80% figure that we have already discussed with Reviewer zfmC?

---

> > > > ### Comment · Reviewer_TYDT · 2024-11-27
> > > >
> > > > I thank the authors for the clarifications. Everything is clear to me, but I encourage the authors to ensure the manuscript reflects previous comments fully and that there is no room for confusion, particularly around distinguishing between real and simulated data.
> > > >
> > > > While I still believe the work primarily reads as a detailed technical report of a specific model that focuses on environmental factors, I recognize that the community could benefit from the practices you propose for detailed reporting of resource consumption at multiple stages of the training/inference pipeline.
> > > > Based on this, I have raised my score to 6. I hope this feedback is helpful, and I look forward to seeing the final version of the paper.

---

### Official Review · Reviewer_zfmC · 2024-11-02

**Soundness:** 4
**Presentation:** 3
**Contribution:** 4
**Rating:** 8
**Confidence:** 4

**Summary:**

This paper estimates the real-world environmental impact of training LLMs, including hardware manufacturing, model development, final training runs, and deployment. They vary model sizes, model architectures, and training time, providing a first view of the embodied CO2 production and water usage from AI systems at these scales. The entire study results in a production of 270t CO2 and a usage of 1.1M liters of water.

Additionally, it finds that model development accounts for 80% of the environmental impact, which is often not stated in related work. Additionally, they find that power draw has highs and lows due to checkpointing, creating a mixed load on the electrical grid, which may not easily be addressed, resulting in control challenges for power providers.

**Strengths:**

S1: This paper provides the first-ever comprehensive end-to-end view of the environmental impact of an LLM. It is highly valuable to have this data published, as we, as a research community, need to have a complete view of how current training and deployment practices affect CO2 production and water usage. The insights and data provided can be used as a building block to argue future performance improvements not only for $ cost reasons but also to quantify their environmental impact.

S2: The authors take care to question current naive assumptions like a 100% power draw during training, making this paper stand out. While this is a low bar, research on environmental impact in LLM training has had its share of invalidated research due to these minor, overlooked details.

S3: The authors estimate the water usage during the development of an LLM, which I have not seen before in this line of research. This adds a new dimension to the environmental impact, providing a more complete picture of how current AI practices affect our environment.

**Weaknesses:**

W1: The result of model development being a large chunk of the environmental impact is not too surprising, but I agree that it is important to track and present in this paper. I am wondering about the representativeness of the data presented in this paper for model development and whether we will see a similar trend continue in the future. Given that this is a key contribution outlined in the abstract, I question whether the number of 80% will change significantly in future related work and if there are steps to take to present this more confidently. I am afraid that researchers in related fields take the final training costs and multiply them by 5x due to the results in this paper.

W2: The second point of discussion has a similar issue as W1. While I agree that oscillating power draw may be a problem for power providers, I hesitate to agree that this is an issue at large. GPU-memory checkpointing has been shown to be possible by Gemini (https://arxiv.org/pdf/2312.11805), which likely reduces this time to sub-seconds. I am not against keeping this insight in general, and that power draw may be an issue for future techniques, but I question the future-proofness of this discussion point. Also, this being a problem for power providers could be explained in more detail and what this implies for the environmental impact.

W3 (minor issues):
* The EU AI Act could be included in 5.1 as it also includes the environmental impact of AI systems (e.g. Art 95 https://artificialintelligenceact.eu/article/95/)
* Figure 1 takes a lot of space for the amount of information it provides. A double y-axis may not be the best visualization as it makes it harder to initially grasp the information. Maybe using two scatter plots would make the visualization more compact and easier to understand?

**Questions:**

I would like the authors to address W1 and W2, whether they agree or not, and if they do, how they plan to address them.

---

> ### Author Response · Authors · 2024-11-25
>
> Thank you for reviewing our work, and your thoughtful comments and suggestions.
>
> Regarding development costs, we agree that the wording of our ‘80%’ result could be made clearer – we will make sure to clarify that our reported development costs amounted to 80% of the final training run costs, rather than 80% of the total costs, such that in our case, the total costs were 1.8x (not 5x) the final runs’ costs – we hope that with more careful wording, people will not naively apply a 5x multiplier to other organizations’ reported final run costs. Though we do not go as far as to claim that our 80% figure is generalizable, we believe that it is a reasonable and roughly representative (conservative) measurement that we hope can inform other researchers and practitioners who set out to develop their own models.
>
>
> With regards to the oscillating power draw, we agree that there are methods of improving training infrastructure to improve the overall process of saving checkpoints, and ideally reducing the impact of major fluctuations in power consumption (though, we note that while GPU-memory checkpointing may reduce the amount of time taken to save a checkpoint, it does not remove it entirely; to do so would instead require a fully asynchronous checkpointing method). However, even some of the largest and best resourced model developers still encounter problems with power fluctuations during checkpointing. For example, the Llama 3 report (https://arxiv.org/pdf/2407.21783, page 14) states:
>
> > *During training, tens of thousands of GPUs may increase or decrease power consumption at the same time, for example, due to all GPUs waiting for checkpointing or collective communications to finish, or the startup or shutdown of the entire training job. When this happens, it can result in instant fluctuations of power consumption across the data center on the order of tens of megawatts, stretching the limits of the power grid. This is an ongoing challenge for us as we scale training for future, even larger Llama models.*
>
> That being said, the reviewer raises a good point, and we will make this argument more clearly in the final version, and more explicitly advocate for improved public model training infrastructure, so the broader ecosystem of smaller developers can take advantage of techniques such as GPU-memory checkpointing (we will also include your recommended citation in the final version).
>
> Regarding your final suggestions, thank you for pointing out that very relevant section of the EU AI Act. We will be sure to include this in the final version. We will also take your suggestions into account for Figure 1; we were planning to make a few improvements to it already, and I believe your suggestions will also help improve it.

---

> ### Comment · Reviewer_zfmC · 2024-11-25
> **Reviewer zfmC Answer to the Authors**
>
> Dear authors,
>
> Thank you for considering my proposals to improve the paper!
>
> Regarding W1), thanks for that clarification of my initial misunderstanding of 1.8x rather than 5x of the cost. Improving the wording will definitely help, and I'm eager to read the updated version. Please take care to frame the results with the exact confidence you have and showcase the limitations in your assumptions such that this results becomes as future-proof as possible.
>
> Regarding W2), thanks for including the reference and the work on GPU-memory based checkpointing. Again, I only hesitantly agree with the way this argument is currently structured. While Meta's Llama 405B is a solid argument for now (good engineers, large investments, etc.), I very much expect this to be one the last models using checkpointing in such a fashion (at least with this kind of frequency). With the way DL progresses right now, this might be outdated by next year and it would be shame if one of the paper's key conclusions becomes obsolete.
>
> Let me outline how I would argue this for the near-term future proof-ness:
> 1) Memory-based checkpointing does not mitigate the risk of catastrophic failures (rack-wide energy loss, earthquakes, spine-level networking equipment breaking down, basically any kind of correlated failures).
> 2) Checkpointing is not only for saving progress to account for failures, but also to restart training in case updates are needed (e.g., updating the learning rate based on some signal).
> 3) Therefore, even with memory-based checkpointing, this does not absolve the need for storage-based checkpointing. Even if live changes to the training algorithm become the norm (2), issue (1) is likely to not solve-able without some kind of progress saving.
>
> For the long-term future, this is still looking pretty rough, as asynchronous checkpointing can become the norm soon, especially if it is solely used to limit the impact of (1). Check out the future work section by IBM + PyTorch here https://pytorch.org/blog/reducing-checkpointing-times/ from June 2024.
>
> All in all, this is a solid contribution for today's checkpointing standards if explained in more detail as the authors suggested. I still would love some more information on how this is an issue for the power providers and why this it is hard for them to account for spikes that could be announced beforehand, but maybe this is a bit out of scope of this conference's target audience. Feel free to incorporate my proposed line of argumentation in your final version or showcase that I am wrong here.
>
> Edit: I've changed my score to 8 (accept, good paper).

---

> > ### Author Response · Authors · 2024-11-26
> >
> > Thank you for your actionable suggestions! They are very helpful.
> >
> > W1: Thank you, we will make sure the final version makes this point clearly, and we will include a discussion of our confidence in the generalizability of our results with regards to model development costs.
> >
> > W2: Thank you for outlining your argument! We will incorporate your line of reasoning in the final version. We would also like to add that even as checkpointing improves, there are other potential causes of large-scale power demand changes, such as waiting for collective communications to finish, as noted by the Llama report. Taken together, these highlight the need for continued improvements in both private and publicly available model training infrastructure, to ensure that solutions to these potentially common problems are able to be used by the broader community.
> >
> > Thank you as well for raising your score!

---

### Official Review · Reviewer_pAyW · 2024-11-03

**Soundness:** 3
**Presentation:** 3
**Contribution:** 3
**Rating:** 8
**Confidence:** 3

**Summary:**

This paper provides estimates and insights into power and water consumption during training and inference of Large Language Models (LLMs). Many of these estimates are based on real experiments in training these models of different parameter sizes. Some are rough estimates for cases where experiments were not possible, such as GPU manufacturing. The paper highlights that there are side activities such as data center cooling, development of the model etc. which are not accounted for in the literature reporting carbon emissions from model training.

**Strengths:**

1. The paper raises very important concerns about transparency in the area of energy and water consumption required for developing as well as using LLMs.
2. This paper includes aspects of this process which have not been reported before such as hyper-parameter tuning and manufacturing of GPUs.
3. The discussion around these calculations is useful for others to understand the environmental implications of doing AI research.

**Weaknesses:**

1. While some of the calculations are clear and seem reproducible as long as some of the manufacturer specific quantities are known, I am not 100% certain if all the steps can be followed by others. It would be useful if the authors can confirm whether someone can apply their methodology for similar experiments/calculations and if the paper contains all the details needed to do so.
2. It would be helpful if the `Development` part of Section 4.1 can provide more details of what it covers i.e. what kind of HPO, what was the search space, what scaling experiments etc.

Minor:
more transparency from developers on when, where, and how they a;re training their models -> more transparency from developers on when, where, and how they are training their models

**Questions:**

I would think similar studies are done for other fields such as electric vehicles. Are there better regulations for reporting in those? What are some other parallels and what other gaps can we find in transparency compared to them?

---

> ### Author Response · Authors · 2024-11-25
>
> Thank you for reviewing our paper, and for your thoughtful comments.
>
> Regarding reproducibility, to the best of our knowledge all of our steps are fully reproducible. We will additionally be releasing our power usage time series data in the final version of the paper, so others can analyze it as well. Other model developers are also able to replicate our calculations for their own training runs, as long as they log power data during training (as this data will be lost otherwise), and if they record where their training runs took place. That being said, we would like to emphasize that the PUE and WUE coefficients we use are specific to our own server hardware and power providers, and anyone trying to report similar numbers for their own LLM development would likely, in the present time, need to reach out individually to their data center provider, etc. This is one aspect of reporting LLM creation costs that we describe in our Discussion section as being a notable challenge in a broader quest for accurate estimation and management of resource consumption driven by LLM creation and use.
>
> Regarding the helpfulness of additional details in the Development section, we agree and will provide further details in the final version of the paper. We are currently withholding some specific information on our development process in order to preserve anonymity, but we will happily provide as much information as possible in the final version, including changes in our model architecture, scaling law and mid-training experiments, and others.
>
> Regarding your question, we agree that an analysis of the gap between reporting requirements in machine learning and other resource-intensive fields would be quite valuable. While we do not have a satisfactory answer to your question now, it is an important point and we will explicitly note this in the paper as a promising area of future research.

---

> > ### Comment · Reviewer_pAyW · 2024-11-26
> >
> > I thank the authors for their response.

---

### Meta-Review · Area_Chair_h3Y6 · 2024-12-19

**Metareview:**

The paper evaluates the end-to-end environmental impact of training LLMs, including hardware manufacturing, model development, final training runs, and deployment. The study accounts for a range of parameters. The take aways are substantiated experimentally. Raising awareness of the conclusions is of general importance and sharing the results will benefit the community. As indicated by one of the reviewers, it is highly valuable to have this data published. The reviewers also suggested a number of ways to improve the manuscript. I would encourage the authors to incorporate these elements in the final version of the paper as it will further strengthen it.

**Additional Comments On Reviewer Discussion:**

All reviewers found the additional clarifications provided by the authors informative. No concerns remained. None of the insights resulting from this work were deemed controversial and found useful to share with the community.

---

### Decision · Program_Chairs · 2025-01-22

Accept (Spotlight)